# An interpretable machine learning model of cross-sectional U.S. county-level obesity prevalence using explainable artificial intelligence

**Ben Allen** *

Department of Psychology, University of Kansas, Lawrence, Kansas, United States of America

* benallen@ku.edu

## Abstract

### Background

There is considerable geographic heterogeneity in obesity prevalence across counties in the United States. Machine learning algorithms accurately predict geographic variation in obesity prevalence, but the models are often uninterpretable and viewed as a black-box.

### Objective

The goal of this study is to extract knowledge from machine learning models for county-level variation in obesity prevalence.

### Methods

This study shows the application of explainable artificial intelligence methods to machine learning models of cross-sectional obesity prevalence data collected from 3,142 counties in the United States. County-level features from 7 broad categories: health outcomes, health behaviors, clinical care, social and economic factors, physical environment, demographics, and severe housing conditions. Explainable methods applied to random forest prediction models include feature importance, accumulated local effects, global surrogate decision tree, and local interpretable model-agnostic explanations.

### Results

The results show that machine learning models explained 79% of the variance in obesity prevalence, with physical inactivity, diabetes, and smoking prevalence being the most important factors in predicting obesity prevalence.

### Conclusions

Interpretable machine learning models of health behaviors and outcomes provide substantial insight into obesity prevalence variation across counties in the United States.

**Data Availability Statement:** All relevant data for this study are publicly available from the OSF repository (https://osf.io/xtfjk/).

**Funding:** The author(s) received no specific funding for this work.

**Competing interests:** The authors have declared that no competing interests exist.

## 1. Introduction

Identifying the principal factors that impact health is an important theme in obesity research [1–3]. Multiple health behaviors and environmental conditions contribute to the obesity crisis [4]. There is also substantial geographic heterogeneity in the prevalence of obesity across the United States [5–7]. Machine learning may be the most powerful approach to modeling variation in obesity prevalence across the United States, but machine learning models are often opaque and difficult to interpret [8]. To open the black box of machine learning models, the field of explainable artificial intelligence has emerged with the goal of extracting domain knowledge about the outcomes being predicted [4, 9–12]. This paper shows an application of explainable artificial intelligence methods to machine learning models of geographic variation in obesity prevalence.

Understanding what machine learning models discover about obesity has the potential to inform public health strategies that address the obesity crisis. Here, an explainable artificial intelligence approach applied to the County Health Rankings data from 2022 helps to better understand the most important factors contributing to the heterogeneity in county-level obesity prevalence [13]. This paper employs four explainable artificial intelligence approaches: 1) random forest estimates of feature importance. 2) Accumulated effects plots that visualize the direction and nature of the discovered associations. 3) A surrogate decision tree trained to mimic the predictions from the random forest model that offers a visual aid in interpreting what the random forest model has learned. 4) Local interpretable model-agnostic explanations offer explanations of obesity prevalence predictions for individual counties. These four explainable artificial intelligence approaches have the potential to leverage the power of machine learning models while extracting information about the important factors contributing to the obesity crisis.

## 2. Methods

### 2.1 Data sources

This paper follows the reporting guidelines for cross-sectional studies outlined by the Strengthening the Reporting of Observational Studies in Epidemiology [14, S2 File]. The analyses in this paper are based on data from the 2022 County Health Rankings [13, 15]. The County Health Rankings dataset is an aggregation of statistics relevant to health for 3,142 counties across the United States. Analysis of publicly available and unidentifiable data does not require approval from the institutional review board. S1 Table contains the sources of all analyzed variables. County-level obesity prevalence based on a body mass index of $\geq 30$ is the predicted outcome in all analyses. The County Health Rankings dataset calculates obesity prevalence using self-reported height and weight from the Behavioral Risk Factor Surveillance System [16]. The predictors used from the county health rankings data include 64 variables from 7 broad categories: health outcomes, health behaviors, clinical care, social and economic factors, physical environment, demographics, and severe housing conditions.

### 2.2 Data pre-processing

R version 4.2.1 (2022-06-23) facilitated all analyses. S1 File contains the Rscript used for all analyses. The analyses omit variables with over 10% missing data. For variable pairs with a Spearman correlation $> \pm 0.90$ (e.g., premature death rate vs. age-adjusted premature death rate), the analysis keeps one variable and omits the other. The analysis also omits values in the County Health Rankings dataset marked as unreliable. The remaining data comprised 65 variables from 3,142 counties.

Data analysis follows a stratified 2-fold cross-validation partitioning scheme for model training and evaluation. The groupdata2 R package (version 2.0.2) divided the full dataset into two partitions (1st partition: 1,570 counties, 2nd partition: 1,572 counties). The partitioning scheme equally balances the two partitions according to the primary outcome (county-level obesity prevalence) and the number of counties from each state.

To estimate missing values for each data partition separately, the multivariate imputation by chained equations R package (M.I.C.E. version 3.14.7) performs 10 imputations with 100 iterations [17]. The final analysis uses the median imputed values. Trace lines of means and standard deviations across iterations showed convergence for each variable. Prevalence of adults with obesity, the primary outcome, was not used to impute any variable.

## 2.3 Statistical analysis

The iterative random forest R package (version 3.0.0) was used to build a random forest prediction model of county-level obesity prevalence using a 2-fold cross-validation scheme [9]. The iterative random forest was used instead of the original random forest algorithm because the iterative approach arrives at more stable estimates of feature importance and provides a more accurate random forest model [9].

First, the modeling algorithm generates a forest of 1,000 decision trees separately for each data partition. The algorithm generates each decision using a subset of 8 features ($\sqrt{64}$ features, the default setting), selected at random from the entire set of 64 features. The algorithm estimates the importance of each feature based on the variance explained in the outcome, averaged across all the decision trees. The algorithm then generates a second prediction model using the same process with one exception: each iteration weights the probability of selecting each feature for a decision tree based on the importance of that feature in the first prediction model. The algorithm iterates 100 times, using the importance from the previously run model, and keeps the model with the best performance based on out-of-bag error. Model performance is quantified using variance explained in the evaluation data (i.e., fold not used in training), as well as the mean absolute difference between the predicted and actual prevalence.

## 2.4 Accumulated local effects

The random forest algorithm estimates the importance of features in predicting obesity prevalence but does not describe the nature of the direction of the relationship. Accumulated local effects plots increase the transparency of what the machine learning model learned about the relationship between individual features and obesity prevalence by showing how the predicted obesity prevalence differs as the value of a feature increases [18]. Subsets of data within specific ranges of feature values are the basis of estimating the accumulated effects. The R package *iml* (version 0.11.1) generated the accumulated local effects plots.

## 2.5 Global surrogate decision tree for random forest model

A global surrogate is an interpretable model trained to simulate the predictions of a machine learning model. The goal is to produce a simple model that provides a general description of how a machine learning model makes predictions. Here, a decision tree is trained on the predictions of a random forest model using the R package *rpart* (version 4.1.16). The tree was grown unrestricted (i.e., complexity = 0) and 10-fold cross-validation was performed to estimate the error for different complexity parameters. To avoid overfitting, the tree was originally pruned using the highest complexity parameter (i.e., 0.001) within 1 standard error of the complexity parameter with the smallest error during cross-validation [19]. However, this resulted in a tree with a depth of nine that was difficult to interpret. Thus, the final surrogate decision

tree was pruned using a more conservative complexity parameter (0.01) to help produce an easily interpreted model.

### 2.6 Local interpretable model-agnostic explanations offer explanations

Whereas the decision tree described above serves as a global surrogate model, the local interpretable model-agnostic explanations approach serves as a local surrogate for individual predictions. The R package *lime* (version 0.5.3) implemented the local interpretable model-agnostic explanations algorithm. The training for each local model uses the prediction model that was not trained on the observation. Interrogation of the local model using the plot_features() function identifies the model features that increase or decrease the predicted prevalence for that county. The main results show the local models for two exemplar counties at lower and higher ends of the obesity prevalence distribution, respectfully.

## 3. Results

Of the 3,142 counties in the analyses, the mean prevalence of adults with obesity was 35.7% (standard deviation = 4.3%; min = 16.4%; max = 51.0%; see Fig 1a). Data partitions served as input for two prediction models for county level obesity prevalence. Each model included 64 features that characterized counties in terms of health outcomes, health behaviors, clinical care, social and economic factors, physical environment, demographics, and severe housing conditions. Implementing a 2-fold cross-validation scheme, the data partition not used in model training is used to evaluate performance. Both models showed similar predictive performance, accounting for a median 79.75% of the variance in the evaluation data ($Model_1$ = 79.77%, $Model_2$ = 79.75%; see Fig 1b). The mean absolute difference between the average predicted prevalence across the two machine learning models and the actual prevalence was 1.4%, with a narrow 95% confidence interval (-0.97%, 3.97%; see Fig 1c and 1d).

### 3.1 Feature importance

Fig 2 shows the 10 most important features, averaged across both prediction models. Feature importance is based on the decrease in residual sum of squares when the decision trees included each of the most important features. The most important feature in the models was physical inactivity, followed by diabetes and adult smoking.

### 3.2 Accumulated local effects

Fig 3A shows a positive linear relationship between physical inactivity and obesity prevalence. Fig 3B shows a similar positive linear relationship between diabetes and obesity prevalence, but with a weaker relationship at lower levels of diabetes prevalence (0.05–0.075). Adult smoking showed a weak positive relationship, with a sharp increase in obesity prevalence at 0.15 (see Fig 3C). There is a negative non-linear relationship between obesity prevalence and the prevalence of uninsured adults, but only the lower end of uninsured adults being associated with higher obesity prevalence (see Fig 3D). The remaining accumulated local effects were relatively flat (see Fig 3E–3J).

### 3.3 Surrogate decision tree model

A surrogate model is an interpretable model that mimics a black-box model's predictions. Here, the interpretable model is a single decision tree trained on the random forest predictions (see Fig 4). The decision tree predictions shared 75.7% of the variance in the random forest predictions and had a mean absolute error of 1.5%. The decision tree predictions shared 65.1%

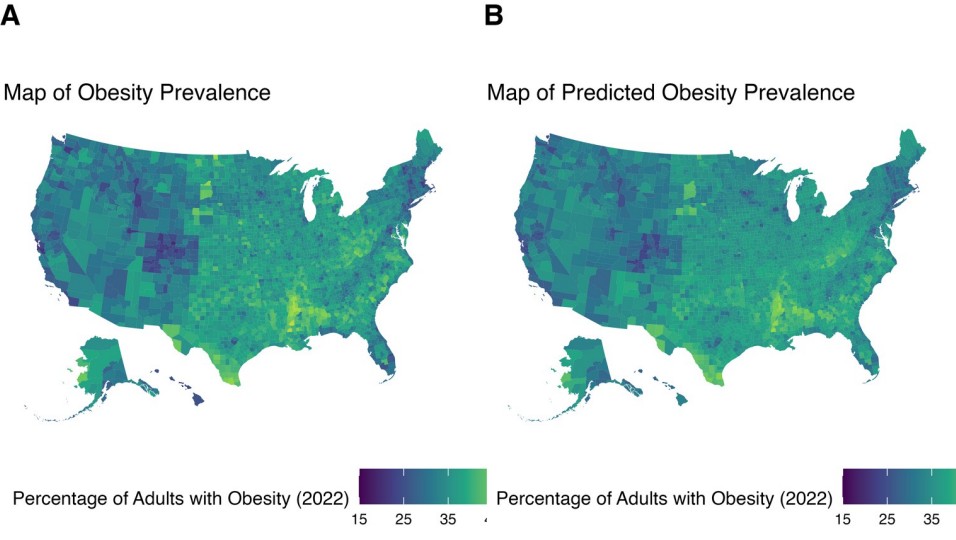

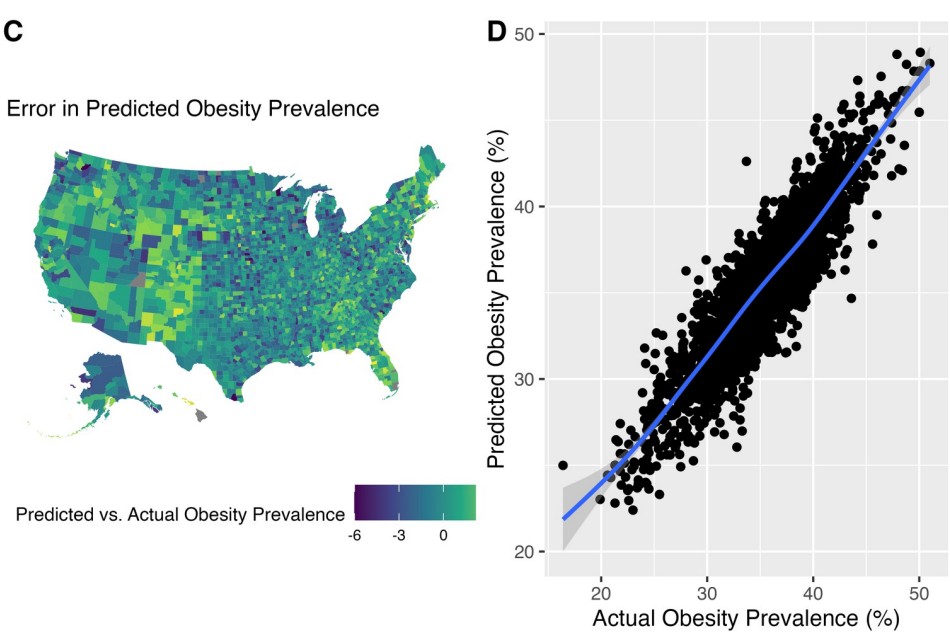

**Fig 1. Maps of actual and predicted obesity prevalence across counties in the United States.** A. Map of actual obesity prevalence; B. Map of obesity prevalence predicted by the random forest models; C. Map of the difference between obesity prevalence and predicted obesity prevalence; D. Scatter plot of predicted and actual obesity prevalence fitted with a generalized additive model function.

of the variance with the actual obesity prevalence values and had a mean absolute error of 1.9%.

The decision tree used 4 of the available 64 features to place each county into one of 9 leaf nodes. Each leaf node corresponded to a specific decision rule (see a complete list of the 9 decision rules in S2 Table). Three of the features included in the decision tree played a prominent role: physical inactivity, diabetes, and smoking prevalence. The prevalence of physical

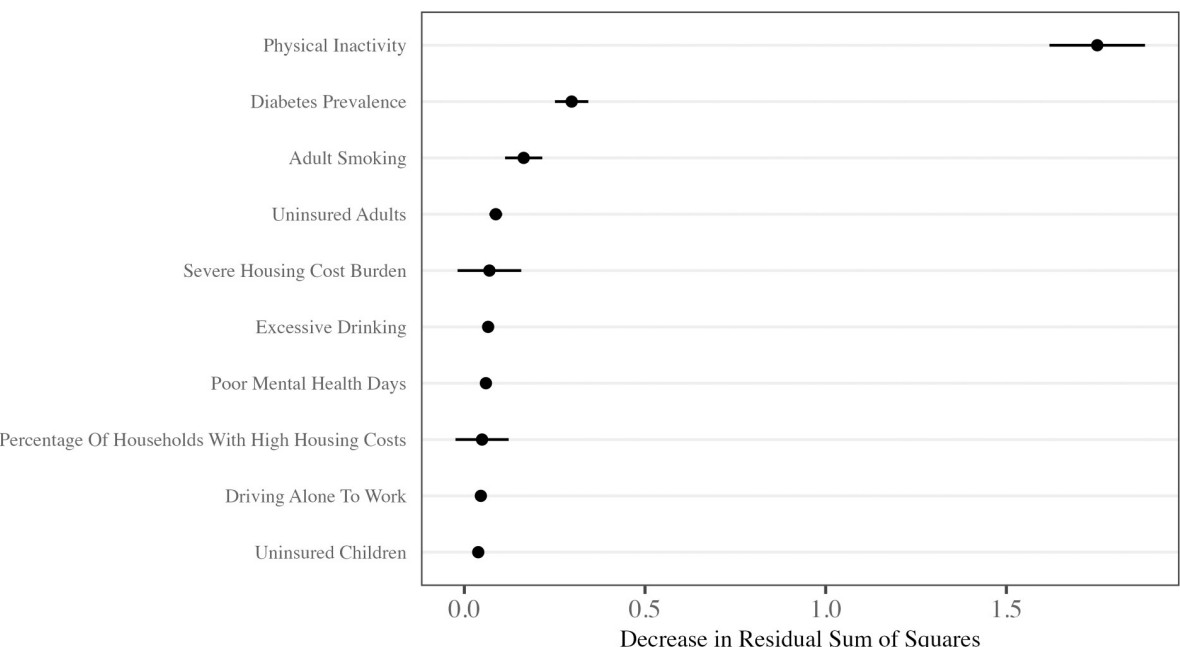

**Fig 2. Ten most important features in predicting county-level obesity prevalence.** Dots show the average decrease in the residual sum of squares decreased when the decision trees included each of the most important features. Lines running through dots extend 2 standard errors. Importance values are averages of the two prediction models.

inactivity dominated the surrogate decision tree, serving as both the root node and nodes on lower branches. Diabetes prevalence has a prominent node on the top of the far right side of the tree that decides the highest levels of obesity prevalence. Conversely, adult smoking prevalence has a prominent node on the top of the far left side of the tree that decides the lowest levels of obesity prevalence. For example, the decision rule for the highest predicted obesity prevalence is: physical inactivity prevalence $> = 0.34$ & diabetes prevalence $> = 0.16$. The decision rule for the lowest predicted obesity prevalence is: physical inactivity prevalence $< 0.18$ & smoking prevalence $< 0.15$.

### 3.4 Local interpretable model-agnostic explanations

Local interpretable model-agnostic explanations offer a way to explain the predicted obesity prevalence for an individual county. To illustrate local models at both high and low ends of the obesity distribution, Fig 5 shows feature plots of the top ten features in the model's prediction for two locations: Dawson County, TX and Benton County, OR. S1 Fig includes plots for all 3,142 counties showing the features that increase or decrease the predicted obesity prevalence.

Dawson County, TX has an obesity prevalence of 0.42, which is at the 96th percentile for obesity prevalence among counties in the United States. The local model predicted the obesity prevalence in Dawson County to be 0.41, with high prevalence of physical inactivity and diabetes being the most important features in predicting the high obesity prevalence. Conversely, Benton County, OR has an obesity prevalence of 0.28, which is at the 5th percentile for obesity prevalence among counties in the United States. The local model predicted the obesity prevalence in Benton County to be 0.27, with low prevalence of physical inactivity and diabetes being the most important features in predicting the low obesity prevalence.

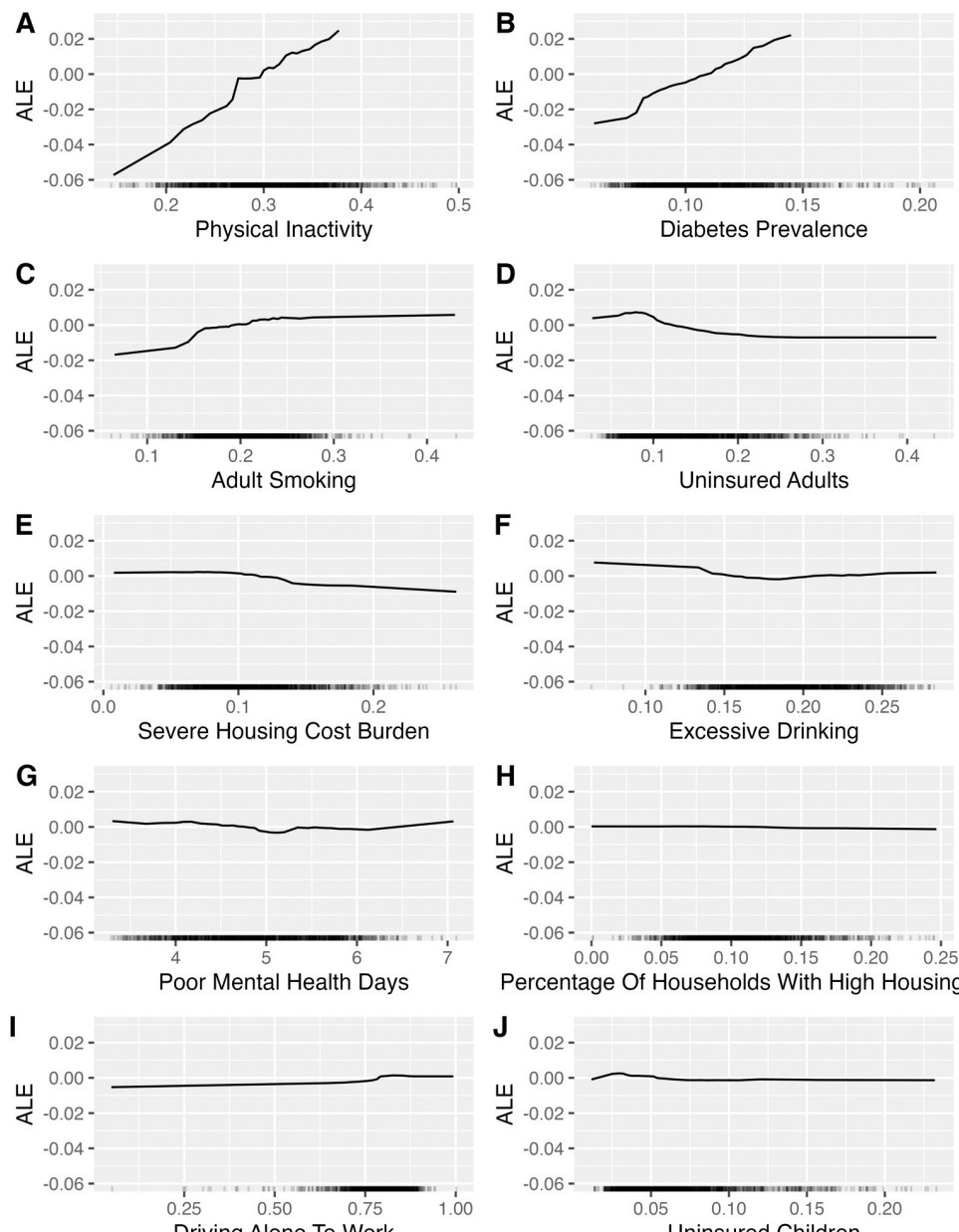

**Fig 3. Accumulated local effects (ALE) of 10 most important features in predicting county-level obesity prevalence.** These plots display the average effect across both prediction models. The y-axis shows how much the model predictions change for subsets of observed values within small ranges of the feature.

## 4. Discussion

This paper shows an explainable artificial intelligence approach to creating an interpretable machine learning model of county-level obesity prevalence. Using a cross-validation approach, two random forest models learned to predict obesity prevalence, and both explained 79% of the heterogeneity in county-level obesity prevalence. Physical inactivity explains most of the heterogeneity in county-level obesity, but the model also highlights the importance of diabetes and smoking.

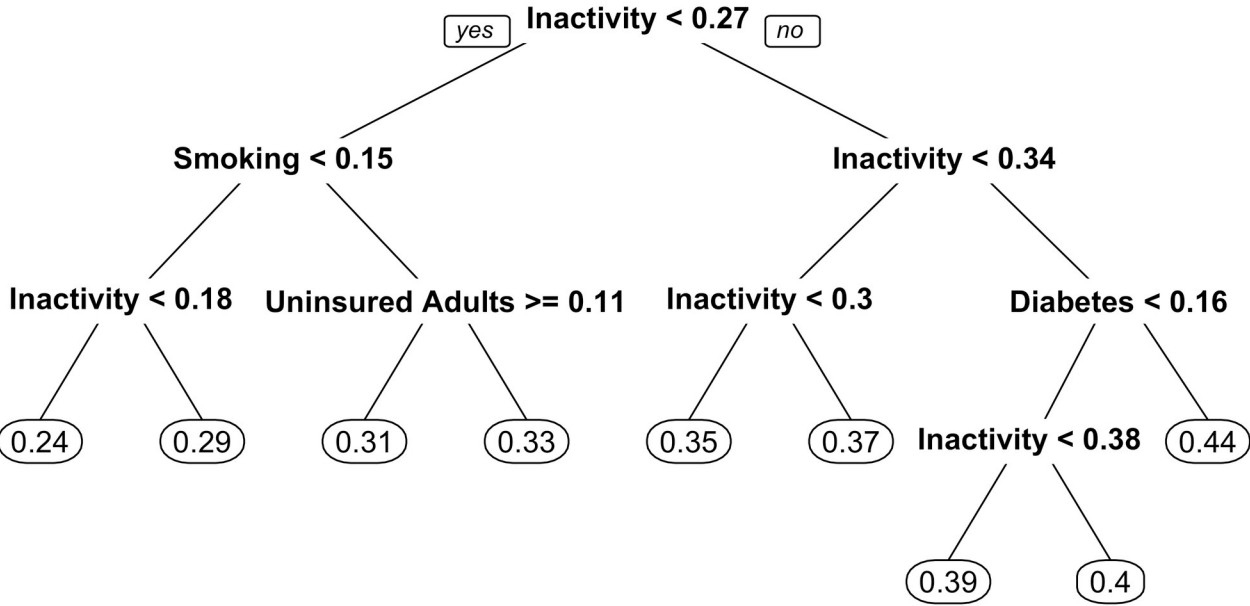

**Fig 4. Global surrogate decision tree.** Decision tree trained on random forest predictions for obesity prevalence. The values in circles at the bottom of the tree are the final predicted obesity prevalence for that branch.

Physical inactivity dominated the random forest models, as well as the global and local surrogate models. The accumulated local effects plot and the example local feature plots suggest a strong linear relationship, with higher levels of physical inactivity linked to higher levels of obesity. While the data analyzed in this paper did not include metrics of energy consumption, higher physical inactivity linked to obesity is consistent with the energy balance model [20, 21]. The analyzed data includes information on the food environment (i.e., food environment index), yet none of these features predicted obesity prevalence. Overall, the data suggest county-level efforts to increase the number of people engaging in monthly physical activities or exercises may be essential to reducing obesity prevalence.

Diabetes played a prominent role in differentiating medium vs. high levels of obesity prevalence in the surrogate decision tree. The accumulated local effects plot showed a linear relationship, with a higher prevalence of diabetes linked to a higher prevalence of obesity. Each diabetes node in the surrogate decision tree showed higher obesity estimates with higher diabetes prevalence. These findings are consistent with extensive research showing obesity causes insulin resistance and can lead to diabetes [22, 23].

Adult smoking played a prominent role in differentiating low vs. medium levels of obesity prevalence in the surrogate decision tree. The accumulated local effects plot showed a weak linear relationship, yet still showing higher prevalence of adult smoking linked to higher prevalence of obesity. The association between higher smoking prevalence and obesity is consistent with evidence that heavy cigarette smoking is associated with higher visceral adiposity and a greater risk of obesity compared to light smokers [24, 25].

The important features in this study differ from previous findings from regression and machine learning models of obesity prevalence using an earlier release of the county health rankings [8]. Aside from using data from a prior year, the authors of this previous study excluded physical inactivity, diabetes, and smoking from their analyses, citing issues of endogeneity in their regression models. In doing so, the authors excluded the most important

**A**

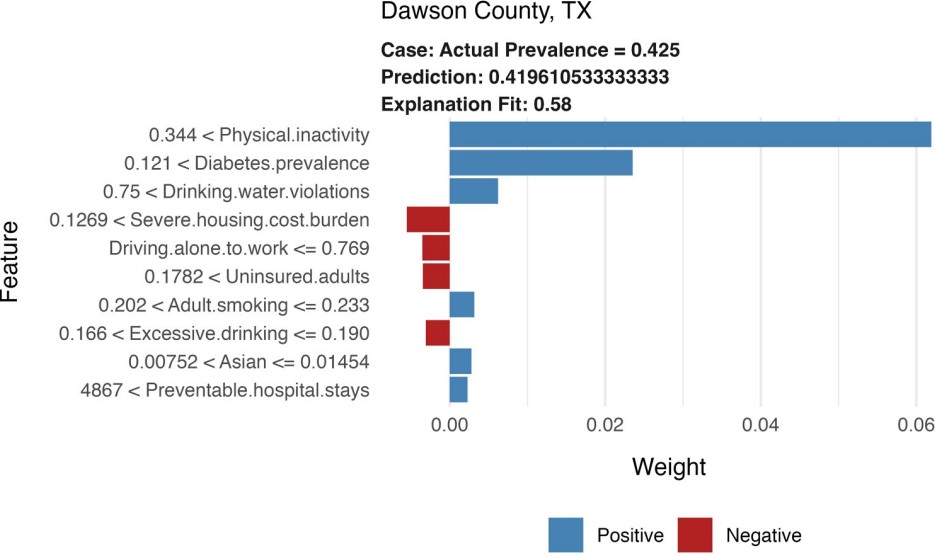

**B**

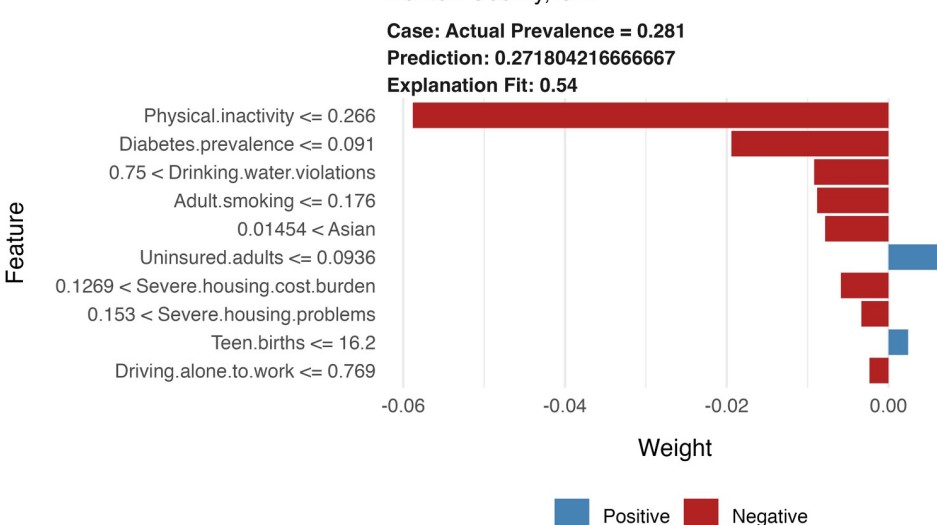

**Fig 5. Feature plots from two local interpretable model-agnostic explanations.** The top of each feature plot shows the actual and local model estimated obesity prevalence. Explanation fit shows the fraction of variance in the local region of the county feature values explained by the local model. The left side of the plot shows the 10 most important features and whether the county was < or <= a threshold value for that feature. Red bars denote features that decrease the predicted obesity prevalence, blue bars denote features that increase the predicted obesity prevalence.

predictors of obesity identified in our analyses, allowing socioeconomic and demographic features to appear more important than they appear in this study.

Thus, the present findings extend earlier machine learning research on county-level obesity prevalence by offering an interpretable machine learning model, not a black box [8]. To the

best of the author's knowledge, this paper is the first using explainable artificial intelligence to build an interpretable machine learning model of county-level obesity prevalence. The iterative random forest approach used in this paper explained 79% of the variance in obesity. The gradient boosting algorithm reported by Sheinker explained only 66%. Though, the lower model accuracy by Sheinker is likely because of the important features omitted in their paper. Here, the results show the power of machine learning models can produce interpretable results through methods from explainable artificial intelligence.

### 4.1 Limitations

Many of the county-level estimates analyzed here are interpolations of self-reported data, sampled from each county. For example, self-reported height and weight are used to estimate obesity prevalence, which introduces error in the primary outcome of the analysis reported here. Moreover, body mass index is an imperfect metric for obesity, compared to waist circumference or skinfold measurements [26]. Yet, the World Health Organization and Centers for Disease Control consider body mass index a reasonable obesity proxy [27, 28].

The cross-sectional data reported here restricts causal claims between features of the model and county-level obesity. This issue is most obvious for diabetes, as obesity is a known risk factor for diabetes, while diabetes is not a risk factor for obesity. Similarly, physical inactivity is a known risk of obesity, yet obesity may reduce the probability of living an active lifestyle. Future studies may show that decreasing physical inactivity prevalence also decreases the prevalence of obesity and diabetes.

Finally, this study does not distinguish between type-1 and type-2 diabetes. Here, diabetes prevalence is based on whether a person self-reports having been told by a doctor they have diabetes. However, type-2 diabetes accounts for 90% to 95% of the diabetes diagnoses in adults, making the findings reported here biased towards resembling a pure measure of type-2 diabetes prevalence [29].

### 5. Conclusion

Explainable artificial intelligence approaches offer the means to increase transparency and interpretability of black-box models, thereby enhancing trustworthiness and ethical oversight of machine learning models in the fields of obesity and biomedicine. By uncovering what these models learn from data, explainable artificial intelligence helps identify crucial factors contributing to obesity, such as physical inactivity, thereby deepening our understanding of the disease. Furthermore, the transparency provided by explainable artificial intelligence facilitates the customization of treatment plans through local interpretable model-agnostic explanations, enabling the identification of specific characteristics and needs of individual counties or patients. Ultimately, explainable artificial intelligence empowers researchers and clinicians to tackle the complexities of obesity, resulting in more effective prevention and treatment strategies.

### Supporting information

**S1 Table. Data sources.**
(PDF)

**S2 Table. Surrogate decision rules.**
(PDF)

**S1 File. Rscript for data analysis.**
(R)

**S2 File. STROBE statement—Checklist of items that should be included in reports of observational studies.**
(DOCX)

**S1 Fig. Local feature plots for all U.S. counties.**
(PDF)

## Author Contributions

**Conceptualization:** Ben Allen.

**Formal analysis:** Ben Allen.

**Investigation:** Ben Allen.

**Methodology:** Ben Allen.

**Project administration:** Ben Allen.

**Writing – original draft:** Ben Allen.

**Writing – review & editing:** Ben Allen.

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
