## [Decision Letter · Decision Letter 0]

3 Jul 2023

PONE-D-23-12550An Interpretable Machine Learning Model of Cross-Sectional U.S. County-Level Obesity Prevalence Rates Using Explainable Artificial IntelligencePLOS ONE

Dear Dr. Allen,

Thank you for submitting your manuscript to PLOS ONE. After careful consideration, we feel that it has merit but does not fully meet PLOS ONE’s publication criteria as it currently stands. Therefore, we invite you to submit a revised version of the manuscript that addresses the points raised during the review process.

We look forward to receiving your revised manuscript.

Kind regards,

Mujeeb Ur Rehman, Ph.D.

Academic Editor

PLOS ONE

2. We note that Figure 1 in your submission contain [map/satellite] images which may be copyrighted. All PLOS content is published under the Creative Commons Attribution License (CC BY 4.0), which means that the manuscript, images, and Supporting Information files will be freely available online, and any third party is permitted to access, download, copy, distribute, and use these materials in any way, even commercially, with proper attribution. For these reasons, we cannot publish previously copyrighted maps or satellite images created using proprietary data, such as Google software (Google Maps, Street View, and Earth). For more information, see our copyright guidelines: http://journals.plos.org/plosone/s/licenses-and-copyright.

Reviewers' comments:

Reviewer's Responses to Questions

**Comments to the Author**

1. Is the manuscript technically sound, and do the data support the conclusions?

Reviewer #1: Partly

Reviewer #2: Yes

2. Has the statistical analysis been performed appropriately and rigorously? 

Reviewer #1: No

Reviewer #2: Yes

3. Have the authors made all data underlying the findings in their manuscript fully available?

Reviewer #1: Yes

Reviewer #2: Yes

4. Is the manuscript presented in an intelligible fashion and written in standard English?

Reviewer #1: Yes

Reviewer #2: Yes

5. Review Comments to the Author

Reviewer #1: • Statistical tests for hypothesis testing and their assumptions should be specified in the study's statistical analysis in the Materials and Methods section.

• The details (version, license number, etc.) of the statistical package(s) or program(s) should be given in the section of "Data Analysis or Statistical Analysis".

• It should be explained how the qualitative and quantitative data are summarized under the sub-heading of Statistical Analyses in the Materials and Methods section of the study.

• Data analysis or Statistical analysis sub-section title should be added to the Materials and Methods.

• The exact P values should be added to the table(s) (e.g., p=0.25; p=0.03).

• Which methods are used to model relationships between variables?

• The descriptions and other descriptive values/data should be defined on the tables and shapes.

• Are the data subjected to pre-processing?

• How were extreme/outlier values in the data determined and resolved?

• What approaches were used to test the validity of the models?

• Which metrics were used in the performance evaluation of the estimates of models/algorithms?

• How were the predictive models selected in this study?

• Which method(s) was/were used to optimize the hyperparameters of models/algorithms?

• How was the most suitable cut-off point determined using the receiver operator characteristic (ROC) curve analysis?

• The number of current references on the subject of the study should be increased.

Reviewer #2: The authors propose an Explainable AI model to predict the Cross-Sectional U.S. county-level obesity prevalence rates. The data was split into 2 folds for training and testing almost 50-50. The dataset has 64 variables from 7 categories. The AI model is built based-on random forest. The manuscript is well-writen and the idea is interesting. However, I have minor concerns/suggestions:

- I lightly surfed the literature and found anothe applications for Explainable AI models in health in general e.g. Covid-19 PMID: 36738712 and for obesity in particular PMID: 35954804. The authors may highlight these types of studies.

- Why the square root of 64. The authors may highlight why decide that or refer to the reference who initially explained this choice.

- How this model avoided over fitting

- AUCROc curve could be added to show the performance of the model.

- the importance of the findings from bio/medical side could be added in the conclusion (2-3 statements).

6. PLOS authors have the option to publish the peer review history of their article (what does this mean?). If published, this will include your full peer review and any attached files.

Reviewer #1: No

Reviewer #2: **Yes: **ABEDALRHMAN ALKHATEEB

<quillbot-extension-portal></quillbot-extension-portal>

---

## [Author Response · Author response to Decision Letter 0]

22 Aug 2023

Thank you for the opportunity to submit a revised manuscript. 

Below are my responses to journal requirements and reviewer comments.

Response to Journal Requirements:

The manuscript should now meet PLOS ONE's style requirements, including those for file naming. 

2. We note that Figure 1 in your submission contain [map/satellite] images which may be copyrighted.

Figure 1 was created for this manuscript by the author using R studio. The data presented in the figures was derived from the models reported in the manuscript as well. I have never published this elsewhere.

3. Please include captions for your Supporting Information files at the end of your manuscript, and update any in-text citations to match accordingly.

Captions for supporting information are now at the end of the manuscript, and in-text citations are updated.

Response to Reviewers' comments:

Reviewer #1: • Statistical tests for hypothesis testing and their assumptions should be specified in the study's statistical analysis in the Materials and Methods section.

The submitted manuscript does not follow a traditional "hypothesis testing" approach. Hypothesis testing requires defining an a priori null hypothesis and an alternative hypothesis. That is not the goal of the submitted paper. Instead, the paper reports the use of machine learning to form a predictive model that performs well on unseen data, that is then probed for knowledge discovery. This process is already described in the Methods section. 

• The details (version, license number, etc.) of the statistical package(s) or program(s) should be given in the section of "Data Analysis or Statistical Analysis".

These were already provided in the manuscript. Section 2.2 Data Pre-Processing shows the version of R used (R version 4.2.1 (2022-06-23)), and the version of each statistical package (i.e., the groupdata2 R package (version 2.0.2)).

• It should be explained how the qualitative and quantitative data are summarized under the sub-heading of Statistical Analyses in the Materials and Methods section of the study.

The paper reports no qualitative data. The only summary statistics reported for quantitative data are means, standard deviations, minimum and maximum values, which require no further explanation.

• Data analysis or Statistical analysis sub-section title should be added to the Materials and Methods.

The paper already has a sub-section titled: 2.3 Statistical Analysis

• The exact P values should be added to the table(s) (e.g., p=0.25; p=0.03).

The manuscript does not report any p-values in a table, so adding exact p-values is impossible. 

• Which methods are used to model relationships between variables?

The manuscript reports in section 2.2 Data Pre-Processing that Spearman correlations were used to examine relationships between variables. But this was only part of pre-processing the data, not a part of the main analyses.

• The descriptions and other descriptive values/data should be defined on the tables and shapes.

The only tables in the manuscript show the data sources and the decision rules, no descriptive values, making this comment difficult to understand. All figures have descriptive captions.

• Are the data subjected to pre-processing?

The pre-processing of the data is described in section 2.2 Data Pre-Processing

• How were extreme/outlier values in the data determined and resolved?

All of the reported data is county-level averages from public data sets that have already been heavily cleaned. Any extreme values in the data are indicative of extreme reality, and were not 'resolved'. Nonetheless, the random forest approach utilized in the manuscript is robust to outliers since they get averaged out by the aggregation of multiple tree output. 

• What approaches were used to test the validity of the models?

The main approach used to test validity was cross-validation described in section 2.2 Data Pre-Processing. Moreover, the validity of the machine learning models were assessed based on their ability to accurately predict obesity values in counties not used in training the model.

• Which metrics were used in the performance evaluation of the estimates of models/algorithms?

Model performance was quantified using variance explained in the evaluation data (i.e., fold not used in training), as well as the mean absolute difference between the predicted and actual prevalence rate. This is now explicity stated in section 2.3 Statistical Analysis.

• How were the predictive models selected in this study?

Iterative random forest models were selected in this study based on a previous report 8 comparing different machine learning algorithms and showing that tree-based models perform well when predicting county-level obesity prevalence. The iterative approach to tree-based models employed in the submitted manuscript has also shown better performance than traditional random forest models, but also lends itself to explainable methods as well. 9 Both of these papers are cited in the manuscript.

8. Scheinker D, Valencia A, Rodriguez F. Identification of Factors Associated With Variation in US County-Level Obesity Prevalence Rates Using Epidemiologic vs Machine Learning Models. JAMA Netw Open. 2019;2(4):e192884. doi:10.1001/jamanetworkopen.2019.2884

9. Iterative random forests to discover predictive and stable high-order interactions | PNAS. Accessed March 30, 2023. https://www.pnas.org/doi/abs/10.1073/pnas.1711236115

• Which method(s) was/were used to optimize the hyperparameters of models/algorithms?

Hyperparamters were not optimized for the iterative random forest models. The creators of the iterative random forest approach have demonstrated that prediction accuracy is fairly robust to hyperparameter choices.9 

9. Iterative random forests to discover predictive and stable high-order interactions | PNAS. Accessed March 30, 2023. https://www.pnas.org/doi/abs/10.1073/pnas.1711236115

• How was the most suitable cut-off point determined using the receiver operator characteristic (ROC) curve analysis?

The paper does not include a receiver operator characteristic (ROC) curve analysis. That analysis is used primarily for evaluating the performance of binary classifiers, where the outcome variable is categorical with two classes. The outcome variable in the submitted manuscript is continuous, making the ROC inappropriate.

• The number of current references on the subject of the study should be increased.

The number of references has been increased based on specific references mentioned by Reviewer 2.

Reviewer #2: The authors propose an Explainable AI model to predict the Cross-Sectional U.S. county-level obesity prevalence rates. The data was split into 2 folds for training and testing almost 50-50. The dataset has 64 variables from 7 categories. The AI model is built based-on random forest. The manuscript is well-writen and the idea is interesting. However, I have minor concerns/suggestions:

- I lightly surfed the literature and found anothe applications for Explainable AI models in health in general e.g. Covid-19 PMID: 36738712 and for obesity in particular PMID: 35954804. The authors may highlight these types of studies.

Thank you for the suggested citations. These papers are now cited in the introduction.

- Why the square root of 64. The authors may highlight why decide that or refer to the reference who initially explained this choice.

This is simply the default parameter for random forest models with a continuous outcome. This is now noted in the method section.

- How this model avoided over fitting

The main strategy used to avoid over-fitting was cross-validation. Model performance was based on the accuracy of predictions on data not used in training. A second strategy was to use a large number of trees (i.e., 1000) in the random forest models. Increasing the number of trees in the forest allows for a more robust aggregation of predictions and reduces the impact of individual noisy trees. 

- AUCROc curve could be added to show the performance of the model.

The AUC-ROC (Area Under the Receiver Operating Characteristic) curve is a commonly used metric for evaluating the performance of binary classification models. However, the submitted manuscript employs a prediction model of a continuous outcome, making the AUC-ROC inappropriate.

- the importance of the findings from bio/medical side could be added in the conclusion (2-3 statements).

The conclusion has been revised to better reflect the importance of the findings and implications for biomedicine research.

---

## [Decision Letter · Decision Letter 1]

7 Sep 2023

PONE-D-23-12550R1An Interpretable Machine Learning Model of Cross-Sectional U.S. County-Level Obesity Prevalence Rates Using Explainable Artificial IntelligencePLOS ONE

Dear Dr. Allen,

Thank you for submitting your manuscript to PLOS ONE. After careful consideration, we feel that it has merit but does not fully meet PLOS ONE’s publication criteria as it currently stands. Therefore, we invite you to submit a revised version of the manuscript that addresses the points raised during the review process.

We look forward to receiving your revised manuscript.

Kind regards,

Mujeeb Ur Rehman, Ph.D.

Academic Editor

PLOS ONE

Journal Requirements:

Reviewers' comments:

Reviewer's Responses to Questions

**Comments to the Author**

1. If the authors have adequately addressed your comments raised in a previous round of review and you feel that this manuscript is now acceptable for publication, you may indicate that here to bypass the “Comments to the Author” section, enter your conflict of interest statement in the “Confidential to Editor” section, and submit your "Accept" recommendation.

Reviewer #2: All comments have been addressed

Reviewer #3: (No Response)

2. Is the manuscript technically sound, and do the data support the conclusions?

Reviewer #2: Yes

Reviewer #3: Partly

3. Has the statistical analysis been performed appropriately and rigorously? 

Reviewer #2: Yes

Reviewer #3: N/A

4. Have the authors made all data underlying the findings in their manuscript fully available?

Reviewer #2: Yes

Reviewer #3: Yes

5. Is the manuscript presented in an intelligible fashion and written in standard English?

Reviewer #2: Yes

Reviewer #3: Yes

6. Review Comments to the Author

Reviewer #2: Te authors have done fair efforts to address the comments. The manuscript is in a very good shape now.

Reviewer #3: This is a revised version of a manuscript that proposes applying iterative RF to predict, in an interpretable fashion, obesity prevalence rates at county level. The study is sound overall, and the prior critique for the most part has been addressed satisfactorily, notably improving the manuscript. Minor issues remain:

1. Why the iterative RF? Ref 8 suggests decision tree-based approaches in general (and gradient boosting in particular), not iterative RF. Regardless, iterative RF is a relatively novel method, and it is not clear to this reviewer that it will perform better than, say, well-established RF + SHAP in this context.

2. Section 2.5 is laconic to the point of incomprehensibility. The opening sentence feels orphaned, the closing sentence is difficult to parse --- what parameters? Hyperparameters? Branching order? Threshold values?

3. Relatedly, just how robust are the splits at the bottom of the tree in Figure 4? Is there any point in growing the tree that deep?

4. In general, overfitting concerns were raised by the prior reviewer(s). This reviewer agrees that growing a large forest of trees addresses the issue somewhat (although higher robustness does not necessarily mean less overfitting/bias); however, the unpruned surrogate tree (Figure 4) and the local explainers (Figure 5) look overfit. Likewise, in Figure 3, it is difficult to quantify just how significant the lower-ranking features are. Yes, this is not a "conventional statistical analysis with p-values" study --- but this is no excuse. How does the reader know whether, for example, the effect in Figure 3 I is significant or not? Why top ten variables in the first place, what was the motivation behind the cutoff?

7. PLOS authors have the option to publish the peer review history of their article (what does this mean?). If published, this will include your full peer review and any attached files.

Reviewer #2: **Yes: **Abedalrhman Alkhateeb

Reviewer #3: No

While revising your submission, please upload your figure files to the Preflight Analysis and Conversion Engine (PACE) digital diagnostic tool, https://pacev2.apexcovantage.com/. PACE helps ensure that figures meet PLOS requirements. To use PACE, you must first register as a user. Registration is free. Then, login and navigate to the UPLOAD tab, where you will find detailed instructions on how to use the tool. If you encounter any issues or have any questions when using PACE, please email PLOS at figures@plos.org. Please note that Supporting Information files do not need this step.<quillbot-extension-portal></quillbot-extension-portal>

---

## [Author Response · Author response to Decision Letter 1]

15 Sep 2023

Comments to the Author

1. If the authors have adequately addressed your comments raised in a previous round of review and you feel that this manuscript is now acceptable for publication, you may indicate that here to bypass the “Comments to the Author” section, enter your conflict of interest statement in the “Confidential to Editor” section, and submit your "Accept" recommendation.

Reviewer #2: All comments have been addressed

Reviewer #3: (No Response)

2. Is the manuscript technically sound, and do the data support the conclusions?

Reviewer #2: Yes

Reviewer #3: Partly

3. Has the statistical analysis been performed appropriately and rigorously?

Reviewer #2: Yes

Reviewer #3: N/A

4. Have the authors made all data underlying the findings in their manuscript fully available?

Reviewer #2: Yes

Reviewer #3: Yes

5. Is the manuscript presented in an intelligible fashion and written in standard English?

Reviewer #2: Yes

Reviewer #3: Yes

6. Review Comments to the Author

Reviewer #2: Te authors have done fair efforts to address the comments. The manuscript is in a very good shape now.

Thank you for the review

Reviewer #3: This is a revised version of a manuscript that proposes applying iterative RF to predict, in an interpretable fashion, obesity prevalence rates at county level. The study is sound overall, and the prior critique for the most part has been addressed satisfactorily, notably improving the manuscript. Minor issues remain:

1. Why the iterative RF? Ref 8 suggests decision tree-based approaches in general (and gradient boosting in particular), not iterative RF. Regardless, iterative RF is a relatively novel method, and it is not clear to this reviewer that it will perform better than, say, well-established RF + SHAP in this context.

Thank you for emphasizing the lack of explanation for the iterative vs. regular RF. Ref 8 does indeed argue for tree-based solutions. But, it should be noted that the main paper describing the iterative RF approach was published at roughly the same time as Ref 8, which is the main reason ref 8 doesn't utilize iterative RF. The main reason we used iterative random forests is because of research in the past 5 years showing that the iterative approach provides more stable estimates of variable importance and the final random forest model generally performs better than the first iteration. The manuscript now includes justification for the iterative RF approach in the beginning of section 2.3 Statistical Analysis.

2. Section 2.5 is laconic to the point of incomprehensibility. The opening sentence feels orphaned, the closing sentence is difficult to parse --- what parameters? Hyperparameters? Branching order? Threshold values?

Section 2.5 is now revised and provides a rich description of the decision tree surrogate. 

The surrogate decision tree was completely rerun based on the reviewer's next comment.

3. Relatedly, just how robust are the splits at the bottom of the tree in Figure 4? Is there any point in growing the tree that deep?

We thank the reviewer for raising the issue of how deep the surrogate decision tree should grow and the robustness of the splits at the bottom. The goal of the surrogate decision tree is to create an interpretable model that approximates the behavior of the random forest model. We agree that some of the splits at the bottom of the decision tree reflected splits of features with low importance. The main issue, as pointed out by the reviewer, is that the surrogate decision tree was not pruned. The revised manuscript addresses this issue by following the practice of pruning decision trees recommended by Breiman et al., (1984) to prune decision trees. 

Breiman, L., Friedman, J., Olshen, R., and Stone, C. (1984). Classification and regression trees. Statistics/Probability Series. Wadsworth & Brooks/Cole Advanced Books & Software.

In the revised manuscript, the surrogate decision tree is grown unrestricted and then pruned based on the complexity parameter and 10-fold cross-validation estimates of the error of each split (Breiman et al., 1984). The tree was pruned using the highest complexity parameter (shortest decision tree) within 1 standard error of the complexity parameter with the smallest error during cross-validation. 

Unfortunately, this recommended practice resulted in an even deeper tree than the one in the previous manuscript. The highest complexity parameter (shortest decision tree) within 1 standard error of the complexity parameter with the smallest error during cross-validation was 0.001, meaning any split that explained 0.1% of the variance was included. To address this issue, we pruned the original decision tree with a more conservative complexity (0.01), which resulted in a short tree that included the major features of the random forest model and only splits that explained at least 1% of the variance in the outcome. This revision makes the surrogate decision tree do a better job of summarizing the model without including less robust features.

4. In general, overfitting concerns were raised by the prior reviewer(s). This reviewer agrees that growing a large forest of trees addresses the issue somewhat (although higher robustness does not necessarily mean less overfitting/bias); however, the unpruned surrogate tree (Figure 4) and the local explainers (Figure 5) look overfit. Likewise, in Figure 3, it is difficult to quantify just how significant the lower-ranking features are. Yes, this is not a "conventional statistical analysis with p-values" study --- but this is no excuse. How does the reader know whether, for example, the effect in Figure 3 I is significant or not? Why top ten variables in the first place, what was the motivation behind the cutoff?

We appreciate the reviewer raising the issue of overfitting. The concern with overfitting is that a machine learning model may perform well on training data but not generalize to unseen data because the model learned noise or other patterns specific to the training data. Cross-validation is one of the main approaches we used to mitigate overfitting, where the random forest model is evaluated based on unseen data withheld from training. One of the ways we tested for overfitting was looking at the variation in mean variable importance during cross-validation (as shown in Figure 2). Fig. 2 shows the standard error bars are quite small, suggesting the models learned similar patterns across data partitions. 

The reason 10 features are plotted is because the tenth most important feature has a mean variable importance that overlaps with zero when considering the standard error, meaning all remaining variables had an average importance of zero, so they aren't plotted. Likewise, the remaining plots focus on the top ten features. However, the local explainers show that the top ten differ across counties, in that the local explainer may not have the same top ten variables as the global model.

As for overfitting with respect to the interpretable models, the purpose of the explainable methods used in the submitted manuscript is not to predict unseen data but to describe the machine learning model in an interpretable way, to see what is inside. The simpler models (i.e., surrogate decision tree, accumulated local effects, local interpretable model-agnostic explanations) are trained on the predictions of the machine learning model rather than the training data itself. The general goal of these models is to fit the predictions as closely as possible. But there is also the issue of interpretabilty, which came into play with the surrogate decision tree. The surrogate tree was pruned to highlight how the most important features make predictions.

---

## [Decision Letter · Decision Letter 2]

20 Sep 2023

An Interpretable Machine Learning Model of Cross-Sectional U.S. County-Level Obesity Prevalence Using Explainable Artificial Intelligence

PONE-D-23-12550R2

Dear Dr. Allen,

We’re pleased to inform you that your manuscript has been judged scientifically suitable for publication and will be formally accepted for publication once it meets all outstanding technical requirements.

Kind regards,

Mujeeb Ur Rehman, Ph.D.

Academic Editor

PLOS ONE

Additional Editor Comments (optional):

Reviewers' comments:

Reviewer's Responses to Questions

**Comments to the Author**

1. If the authors have adequately addressed your comments raised in a previous round of review and you feel that this manuscript is now acceptable for publication, you may indicate that here to bypass the “Comments to the Author” section, enter your conflict of interest statement in the “Confidential to Editor” section, and submit your "Accept" recommendation.

Reviewer #3: All comments have been addressed

2. Is the manuscript technically sound, and do the data support the conclusions?

Reviewer #3: Yes

3. Has the statistical analysis been performed appropriately and rigorously? 

Reviewer #3: N/A

4. Have the authors made all data underlying the findings in their manuscript fully available?

Reviewer #3: Yes

5. Is the manuscript presented in an intelligible fashion and written in standard English?

Reviewer #3: Yes

6. Review Comments to the Author

Reviewer #3: The authors have satisfactorily addressed the prior criticisms and suggestions. The manuscript is ready for publication.

7. PLOS authors have the option to publish the peer review history of their article (what does this mean?). If published, this will include your full peer review and any attached files.

Reviewer #3: No

<quillbot-extension-portal></quillbot-extension-portal>

---

## [Editor Report · Acceptance letter]

27 Sep 2023

PONE-D-23-12550R2 

An Interpretable Machine Learning Model of Cross-Sectional U.S. County-Level Obesity Prevalence Using Explainable Artificial Intelligence 

Dear Dr. Allen:

I'm pleased to inform you that your manuscript has been deemed suitable for publication in PLOS ONE. Congratulations! Your manuscript is now with our production department. 

Kind regards, 

on behalf of

Dr. Mujeeb Ur Rehman 

Academic Editor

PLOS ONE